# Socioeconomic position is associated with *N-terminal pro-brain natriuretic peptide (NT-proBNP)*—Results of the population-based Heinz Nixdorf Recall study

**Marina Rudman[1], Mirjam Frank[1], Carina Emmel[1], Emanuel Matusch[1], Kaffer Kara[2], Amir Abbas Mahabadi[3], Raimund Erbel[1], Karl-Heinz Jöckel[1], Nico Dragano[4], Börge Schmidt[1] ***

1 Institute for Medical Informatics, Biometry and Epidemiology, University of Duisburg-Essen, Essen, Germany, 2 Department of Cardiology, Agaplesion Hospital Hagen, Hagen, Germany, 3 Department of Cardiology and Vascular Medicine, West German Heart and Vascular Center Essen, University of Duisburg-Essen, Essen, Germany, 4 Institute of Medical Sociology, University Hospital Düsseldorf, Düsseldorf, Germany

* boerge.schmidt@uk-essen.de

**Data Availability Statement:** Due to data security reasons (i.e., data contain potentially participant identifying information), the Heinz Nixdorf Recall

## Abstract

### Objectives

N-Terminal pro Brain Natriuretic Peptide (NT-proBNP) is a diagnostic marker for heart failure and a prognostic factor for cardiovascular disease (CVD). The aim of this study was to examine the association of socioeconomic position (SEP) with NT-proBNP while assessing sex-differences and the impact of CVD risk factors and prevalent CVD on the association.

### Methods

Baseline data of 4598 participants aged 45–75 years of the Heinz Nixdorf Recall Study were used. Income and education were used as SEP indicators. Age- and sex-adjusted linear regression models were fitted to calculate effect size estimates and 95% confidence intervals (95%-CIs) for the total effect of SEP indicators on NT-proBNP, while potential mediation was assessed by additionally accounting for traditional CVD risk factors (i.e., systolic blood pressure, HDL cholesterol, LDL cholesterol, diabetes, anti-hypertensive medication, lipid-lowering medication, BMI, current smoking). Education and income were included separately in the models.

### Results

With an age- and sex-adjusted average change in NT-proBNP of -6.47% (95%-CI: -9.91; -2.91) per 1000€, the association between income and NT-proBNP was more pronounced compared to using education as a SEP indicator (-0.80% [95%-CI: -1.92; 0.32] per year of education). Sex-stratified results indicated stronger associations in men (-8.43% [95%-CI: -13.21; -3.38] per 1000€; -1.63% [95%-CI: -3.23; -0.001] per year of education) compared to women (-5.10% [95%-CI: -9.82; -0.01] per 1000€; -1.04% [95%-CI: -2.59; 0.50] per year of

Study does not allow sharing data as a public use file. However, others can access the data used upon request, which is the same way authors of the present paper obtained the data. Data requests can be addressed to: recall@uk-essen.de.

**Funding:** This work was supported by the Heinz Nixdorf Foundation (RE, KHJ); the German Ministry of Education and Science [grants Nationales Genomforschungsnetz, 01GS0820] (RE, KHJ), the German Research Council [projects SI 236/8-1, SI 236/9-1] (RE, KHJ). The funders had no role in study design, data collection and analysis, decision to publish, or preparation of the manuscript.

**Competing interests:** The authors have declared that no competing interests exist.

education). After adjusting for CVD risk factors some of the observed effect size estimates were attenuated, while the overall association between SEP indicators and NT-proBNP was still indicated. The exclusion of participants with prevalent coronary heart disease or stroke did not lead to a substantial change in the observed associations.

## Conclusions

In the present study associations of education and income with NT-proBNP were observed in a population-based study sample. Only parts of the association were explained by traditional CVD risk factors, while there were substantial sex-differences in the strength of the observed association. Overt coronary heart disease or stroke did not seem to trigger the associations.

## Introduction

Cardiovascular disease (CVD) is the most common cause of death in western countries [1]. It is well known that the incidence of CVD is not equally distributed across social strata with a higher CVD risk in groups of low socioeconomic position (SEP) [2]. Even though there is strong evidence that SEP affects CVD via unequally distributed risk factors such as health-related behaviors (e.g., smoking, nutrition), psychosocial factors (e.g., mental stress) and different material factors (e.g., living, working and housing conditions) [3], the specific pathways linking SEP with CVD risk are not sufficiently understood. Sex-differences in SEP-associated risk factors (e.g., prolonged smoking) as well as sex-specific CVD risks (e.g., common pregnancy disorders such as gestational hypertension/diabetes or common endocrine disorders such as polycystic ovary syndrome) additionally complicate the elucidation of the SEP-CVD association as well as the development of tailored prevention strategies for high risk groups as an important goal for epidemiological research [4].

Exploring the association of SEP with CVD by assessing CVD events only may not sufficiently reflect the distribution of CVD risk in populations. Especially in low SEP groups a substantial proportion of individuals may be undertreated while showing subclinical signs of being at high risk for developing CVD events. The N-terminal pro-Brain Natriuretic Peptide (NT-proBNP), established as the leading diagnostic marker for heart failure, has been described in recent years in numerous population-based study samples as a promising prognostic factor for CVD events [5–7]. For both conditions, heart failure and CVD, sex-differences have also been reported with respect to clinical characteristics, therapeutic responses to treatments and prognosis [8–10]. As NT-proBNP has been reported to be higher in women compared to men in population-based studies [11–13], sex-differences have to be considered when defining NT-proBNP reference ranges or cut-off values for clinical diagnosis of heart failure as well as for CVD prediction. The exploration of the NT-proBNP distribution across SEP groups considering sex-differences is thus of high interests to enhance the understanding of how SEP affects heart failure and CVD risk.

A recent study has given first indication for an inverse association between SEP and NT-proBNP in a population-based cohort excluding participants with prevalent CVD [14]. However, potential sex-differences of the association were not investigated, despite evidence for differences in NT-proBNP and CVD risk factors between men and women [15–17]. The aim of the present study was to examine the association of SEP indicators education and income with

NT-proBNP, while considering possible sex-differences, the potential influence of traditional CVD risk factors and the impact of prevalent CVD on the association. The Heinz Nixdorf Recall study was used as study population, where the association of SEP with CVD risk [15] and the prognostic value of NT-proBNP for future CVD events in healthy subjects [16, 17] have already been reported.

## Methods

### Study population

Data were used from the baseline examination of the Heinz Nixdorf Recall study, which is a population-based cohort study established to investigate the predictive value of novel markers for CVD risk stratification in addition to traditional CVD risk factors. Rationale and design of the study have been described previously [18]. In short, the baseline examination took place from 2000 to 2003, where 4814 women and men (proportion of women: 50.2%) aged 45-75 years were recruited using a random sample from mandatory citizen registries of three large cities (Bochum, Essen, Mulheim/Ruhr) in an urban region in the western part of Germany. The baseline response proportion was 56% [19]. Written informed consent was obtained from all participants. The study was accredited by the institutional ethics committee of the University Hospital Essen and contained extended quality management procedures, including a certification according to DIN ISO 9001:2000.

### Indicators of socioeconomic position

Information on SEP indicators education and income was obtained by standardized face-to-face interviews at study baseline. Income was measured as the monthly household equivalent income calculated by dividing the total household net income by a weighting factor for each household member [20]. Income was used for analysis either as a continuous variable or divided into four groups using sex-specific quartiles. Education was defined according to the International Standard Classification of Education as total years of formal education combining school and vocational training [21]. As in previous analyses of the same study population education was used as a continuous variable or categorized into four different groups with the lowest educational group of 10 and less years (equivalent to a basic school degree with no vocational training), the group of 11 to 13 years (equivalent to upper secondary educational degrees or a combination of lower secondary educational and vocational training), the group of 14 to 17 years (equivalent to a vocational training including additional qualification) and the highest group of 18 and more years of education (equivalent to a university degree) [15].

### NT-proBNP

NT-proBNP was determined from frozen plasma samples collected at study baseline. Immediately after collection, blood samples were centrifugalized, aliquoted and then stored at −80°C. The Roche Modular E170 Assay (Roche Diagnostics, Mannheim, Germany) was used to measure NT-proBNP. The analytic functional sensitivity of the assay representing the lowest NT-proBNP concentration determined was 5 pg/ml.

### Cardiovascular risk factors

Traditional CVD risk factors included in the analysis were collected at study baseline [22]. Three automatic blood pressure measurements were conducted using the oscillometric method (device: Omron HEM-705-CP) [23]. The arithmetic mean of the second and third measurement and information on anti-hypertensive medication was used for analysis. Levels

of high density lipoproteine (HDL), low density lipoprotein (LDL) and total cholesterol were determined using standard enzymatic methods. The blood samples were examined within 12 hours in the central laboratory of the University Hospital Essen according to usual standards. Information on lipid-lowering medication such as statins was also collected. Diabetes was defined as either of the following criteria: reported history of diabetes, taking glucose-lowering drugs, having fasting blood glucose levels greater than 125 mg/dL, or having nonfasting glucose levels of 200 mg/dL or greater [24]. The body-mass-index (BMI) was calculated from standardized measurements of body weight and height and defined as the ratio of body weight in kilograms to the square of body height in meters ($kg/m^2$). Information on smoking was obtained from standardized face-to-face interviews and current smoking was defined as having smoked during the last 12 months [25]. To estimate the glomerular filtration rate (GFR), the MDRD formula (modification of diet in renal disease) was used: ($ml / min / 1.73 \ m^2$) = 186 x serum creatinine$^{-1.154}$ x age$^{-0.203}$ (x 0.742 for women) [16]. A GFR value of 30 ml / min / 1.73 $m^2$ or less was defined as renal insufficiency [16].

## Statistical analysis

Out of the 4814 participants of the Heinz Nixdorf Recall study, 203 participants had no information on NT-proBNP. Additionally, participants with a low GFR (GFR ≤ 30 ml/min/1.73 $m^2$) or missing GFR (n = 13) were excluded, because an impaired renal function has been shown to affect the plasma concentration of NT-proBNP. As there were some observations missing for cardiovascular risk factors, participants were excluded from the respective analyses. Finally, 4598 participants were included in the main analysis population. The effective sample size for multivariate analyses using education as SEP indicator was 4585 (missing values n = 13) and 4305 for using income (missing values n = 293). Subjects with missing NT-proBNP values showed small differences in average income and in the proportions of educational groups compared to the main analysis population. These differences may be explained by the higher proportion of women in the group of subjects with missing NT-proBNP values. The group of subjects with missing information on income had a higher proportion of women and higher NT-proBNP values (data not shown).

In addition to the main analysis population, a subsample for sensitivity analysis was defined by excluding participants with known coronary heart disease (n = 327) and stroke (n = 135)) at study baseline, as these outcomes are known to be associated with high NT-proBNP levels as well as low SEP. Prevalent coronary heart disease was defined as known myocardial infarction, coronary bypass surgery and/or interventional revascularization [18]. The aim of the sensitivity analysis was to assess the impact of prevalent CVD on the investigated association by checking the robustness of the main study results in asymptomatic participants. In total, 4160 subjects were included in the sensitivity analysis using education as SEP indicator (missing values n = 9) and 3898 using income (missing values n = 271).

To characterize the study population, mean ± standard deviation (SD) or median with interquartile range (IQR: Q1; Q3) were reported for continuous variables and absolute numbers (N) and frequency (%) for categorical variables. To assess the association between SEP indicators and NT-proBNP linear regression models were fitted with NT-proBNP as the dependent variable. As NT-proBNP was not normally distributed, it was log-transformed prior to analysis. Education and income were included separately in linear regression models either as continuous or categorical independent variables to check for the robustness of the study results. The beta estimates obtained from the linear regression analysis were back transformed and presented as percentage change in NT-proBNP per unit ([exp(beta)-1]*100). For categorical SEP variables dummies were used with the category ≥18 years of education or the

highest income quartile as reference. Since it has been observed that average NT-proBNP differs by sex [26–28], all analyses were additionally conducted sex-stratified.

Linear regression models were fitted with different adjustment sets. A minimal sufficient adjustment set to estimate the total SEP effect included age and sex (model 1), while the direct effect was estimated additionally adjusting separately for each CVD risk factor (systolic blood pressure, diastolic blood pressure, total cholesterol, HDL cholesterol, LDL cholesterol, diabetes, anti-hypertensive medication, lipid-lowering medication, BMI, smoking) as potential mediators as well as including all CVD risk factors in one model (model 2). Diastolic blood pressure was excluded from model 2 due to its high correlation with systolic blood pressure. Total cholesterol was excluded from model 2 due to its high correlation with LDL. To illustrate the precision of effect size estimates, 95% confidence intervals were calculated. All statistical analyses were performed using R (Version Rx64 3.3.2).

## Results

Characteristics of the study population are shown in Table 1. On average, the main analysis population had an NT-proBNP plasma concentration of 71 pg/ml (IQR: 39; 132). There were higher NT-proBNP median levels observed for women compared to men (women: 86 pg/ml [IQR: 52; 147]; men: 55 pg/ml [IQR: 31; 109]), while men were more exposed to traditional CVD risk factors than women, including marked differences in blood pressure, HDL cholesterol, diabetes mellitus and smoking. The distribution of education and income also differed between women and men with higher income and education reported in men. After excluding participants with prevalent coronary heart disease and stroke the distribution of income and education differed only slightly compared to the main analysis population (S1 Table). The NT-proBNP plasma concentration was on average 4 pg/ml lower compared to the main analysis population. No substantial differences in the distribution of traditional CVD risk factors were observed.

The boxplots in Fig 1 illustrate the crude association of the log-transformed NT-proBNP and categorical SEP indicators. For both, men and women, a downward trend of NT-proBNP was observed across education and income categories with the highest average NT-proBNP levels in the low SEP groups.

Table 2 shows the percentage change of NT-proBNP per 1000€ income in the main analysis population (please see S2 Table for all model effect size estimates). In the age- and sex-adjusted model 1, NT-proBNP decreased by 6.47% (95%-CI: -9.91; -2.91) per additional 1000€/month. After additionally adjusting for traditional cardiovascular risk factors (model 2), the effect size was slightly reduced (-5.75; 95%-CI: -9.25; -2.13). The sex-stratified analyses revealed a stronger association between income and NT-proBNP for men compared to women. In age- and sex-adjusted regression models with additional adjustment for single CVD risk factors, no major change was observed for any of the effect size estimates illustrating the association of income with NT-proBNP (S3 Table).

Using education as SEP indicator, NT-proBNP decreased by 0.80% (95%-CI: -0.32; 1.92) per additional year of education in model 1 (Table 3; please see S4 Table for all model effect size estimates). The effect size estimate changed only slightly after additionally adjusting for traditional cardiovascular risk factors (model 2). Again, for men stronger effect size estimates were observed compared to women. In age- and sex-adjusted regression models with additional adjustment for single CVD risk factors, major changes of the effect size estimates for the association of education with NT-proBNP were not observed (S5 Table). After excluding participants with prevalent coronary heart disease and stroke, effect size estimates for both income and education differed only slightly compared to the main analysis population (S6 and S7 Tables).

Table 1. Characteristics of the main study population stratified by sex (mv: Number of missing values).

| | All | Women | Men |
|---|---|---|---|
| N (%) | 4598 (100%) | 2293 (49.9%) | 2305 (50.1%) |
| Age (years) | 59.6 ± 7.8 | 59.6 ± 7.8 | 59.7 ± 7.8 |
| Income (EURO/month) mv = 293 | 1449 (1108; 1875) | 1313 (937; 1875) | 1520 (1108; 2073) |
| Education (years of training) mv = 13 | | | |
| ≤ 10 (low) | 524 (11.4%) | 410 (17.9%) | 114 (5.0%) |
| 11–13 | 2536 (55.3%) | 1446 (63.1%) | 1090 (47.5%) |
| 14–17 | 1037 (22.6%) | 259 (11.3%) | 778 (33.9%) |
| ≥ 18 (high) | 488 (10.6%) | 175 (7.6%) | 313 (13.6%) |
| NT-proBNP (pg/ml) | 71 (39; 132) | 86 (52; 147) | 55 (31; 109) |
| BMI (kg/m$^2$) mv = 25 | 27.9 ± 4.6 | 27.6 ± 5.2 | 28.2 ± 4.0 |
| Systolic blood pressure (mmHg) mv = 12 | 133.2 ± 20.8 | 128.2 ± 20.9 | 138.1 ± 19.5 |
| Diastolic blood pressure (mmHg) mv = 11 | 81.5 ± 10.9 | 78.9 ± 10.6 | 84.0 ± 10.6 |
| Anti-hypertensive medication mv = 16 | 1623 (35.4%) | 786 (34.3%) | 837 (36.5%) |
| Total cholesterol (mg/dl) | 229.0 ± 39.1 | 233.3 ± 39.3 | 224.8 ± 38.5 |
| LDL cholesterol (mg/dl) mv = 13 | 145.5 ± 36.2 | 145.8 ± 36.7 | 145.3 ± 35.6 |
| HDL cholesterol (mg/dl) mv = 1 | 58.1 ± 17.2 | 65.2 ± 17.0 | 51.1 ± 14.4 |
| Lipid-lowering medication mv = 281 | 564 (13.1%) | 244 (11.3%) | 320 (14.9%) |
| Diabetes mellitus | 617 (13.4%) | 217 (9.5%) | 400 (17.4%) |
| Current Smoking mv = 7 | 1071 (23.3%) | 490 (21.3%) | 581 (25.3%) |

Results of the analyses using categorical SEP indicators also revealed socioeconomic differences in NT-proBNP. Compared to the highest income quartile, the lower income quartile showed an 11.94% (95%-CI: 3.75; 20.77) higher average NT-proBNP level after adjusting for age and sex (Fig 2). In women, the largest difference was observed between the highest and lowest income group, while in men much stronger effect size estimates were observed for all lower income groups compared to the highest. After adjusting for traditional cardiovascular risk factors effect size estimates were less strong in men, while only small changes were observed in women. The exclusion of prevalent coronary heart disease and stroke led to smaller effect size estimates, although strong effects were still observed in men (S8 Table).

Compared to the highest education group, NT-proBNP was on average 7.46% (95%-CI: -3.93; 20.20) higher in the lowest education group (Fig 3). However, the medium education groups differed more strongly from the highest education group in the average NT-proBNP plasma concentration compared to the lowest education group. The addition of traditional cardiovascular risk factors as covariates did not lead to substantial changes in the effect size estimates. For males, stronger group differences between the reference group and the lower education groups were observed compared to women. After excluding prevalent coronary heart disease and stroke, most of the effect size estimates were slightly smaller, while the overall higher NT-proBNP levels in the lower education groups were still observed (S9 Table).

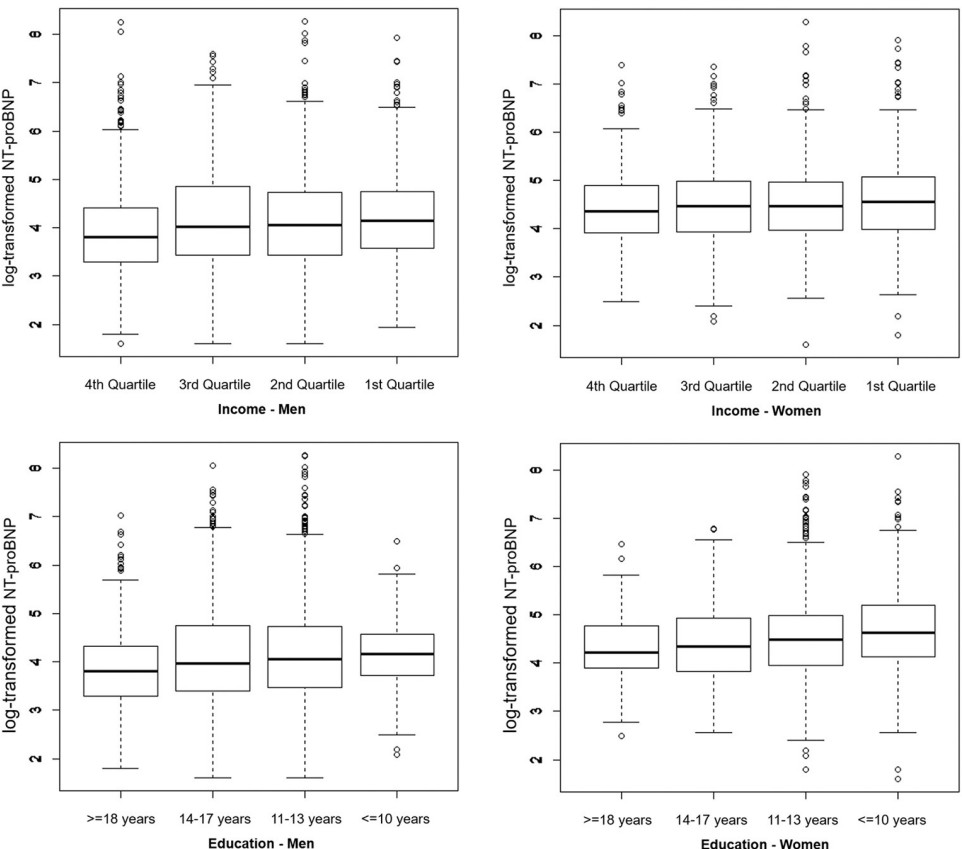

**Fig 1. Boxplots illustrating the distribution of log-transformed NT-proBNP across income quartiles and education groups stratified by sex.**

**Table 2. Effect size estimates as percentage change in NT-proBNP per 1000€ income/month and 95% confidence intervals (95%-CI) for the main analysis population and stratified by sex.**

| All | | | |
|---|---|---|---|
| **Model** | **N** | **%-Change** | **95%-CI** |
| **Model 1** | 4305 | -6.47 | -9.91; -2.91 |
| **Model 2** | 4013 | -5.75 | -9.25; -2.13 |
| **Men** | | | |
| **Model** | **N** | **%-Change** | **95%-CI** |
| **Model 1** | 2208 | -8.43 | -13.21; -3.38 |
| **Model 2** | 2042 | -7.36 | -12.24; -2.21 |
| **Women** | | | |
| **Model** | **N** | **%-Change** | **95%-CI** |
| **Model 1** | 2097 | - 5.10 | -9.82; -0.01 |
| **Model 2** | 1971 | -4.95 | -9.71; 0.07 |

Model 1: Adjusted for age, (sex); Model 2: Adjusted for age, (sex), systolic blood pressure, HDL cholesterol, LDL cholesterol, diabetes, anti-hypertensive medication, lipid-lowering medication, BMI and current smoking.

**Table 3. Effect size estimates as percentage change in NT-proBNP per year of education and 95% confidence intervals (95%-CI) for the main analysis population and stratified by sex.**

| All | | | |
|---|---|---|---|
| **Model** | **N** | **%-Change** | **95%-CI** |
| **Model 1** | 4585 | -0.80 | -1.92; 0.32 |
| **Model 2** | 4274 | -0.85 | -1,99; 0.30 |
| Men | | | |
| **Model** | **N** | **%-Change** | **95%-CI** |
| **Model 1** | 2295 | -1.63 | -3.23; -0.001 |
| **Model 2** | 2125 | -1.44 | -3.08; 0.23 |
| Women | | | |
| **Model** | **N** | **%-Change** | **95%-CI** |
| **Model 1** | 2290 | -1.04 | -2.59; 0.50 |
| **Model 2** | 2149 | -1.24 | -2.79; 0.33 |

Model 1: Adjusted for age, (sex); model 2: Adjusted for age, (sex), systolic blood pressure, HDL cholesterol, LDL cholesterol, diabetes, anti-hypertensive medication, lipid-lowering medication, BMI and current smoking.

## Discussion

In the present study associations of the SEP indicators education and income with NT-proBNP were observed in a population-based study sample aged 45-75 years. The association between income and NT-proBNP was more pronounced compared to using education as a SEP indicator. Sex-stratified differences indicated stronger socioeconomic differences in NT-proBNP for men. After adjusting for traditional cardiovascular risk factors, some of the observed effect size estimates were attenuated, while the overall association between SEP indicators and NT-proBNP was still indicated.

In one previous study investigating the association between SEP and NT-proBNP in a population-based sample, Vart et al. (2018) have examined the SEP indicators educational attainment and household income regarding the level of NT-proBNP [14]. The study population consisted of 12,646 participants from the ARIC study after excluding prevalent cardiovascular events (coronary heart disease, stroke and hospitalization due to heart failure). A social gradient has been detected in NT-proBNP for both, education and income. Overall, our results were in line with this recently published study. Results of Vart et al. (2018) have also indicated that the association between SEP and NT-proBNP did not seem to be completely mediated by traditional cardiovascular risk factors. This is again in line with our results, where the association of SEP indicators and NT-proBNP was not strongly reduced after adjusting for traditional CVD risk factors (systolic blood pressure, HDL cholesterol, LDL cholesterol, diabetes, anti-hypertensive medication, lipid-lowering medication, BMI, current smoking). The small explanatory power of traditional cardiovascular risk factors for the observed association between SEP and NT-proBNP may be explained by mediation through other factors not considered so far.

As the overall NT-proBNP plasma concentration was higher in women, sex-stratified analyses indicated that men showed larger SEP differences in NT-proBNP compared to women. While sex-differences in CVD-related health inequalities have in some study populations been attributed to varying distributions of cardiovascular risk factors between sexes [2, 29], the small explanatory power of the cardiovascular risk factors included in the analysis was observed in both, women and men. Thus, observed sex-differences in the prevalence of CVD risk factors such as diabetes mellitus or hypercholesterolemia are probably not responsible for

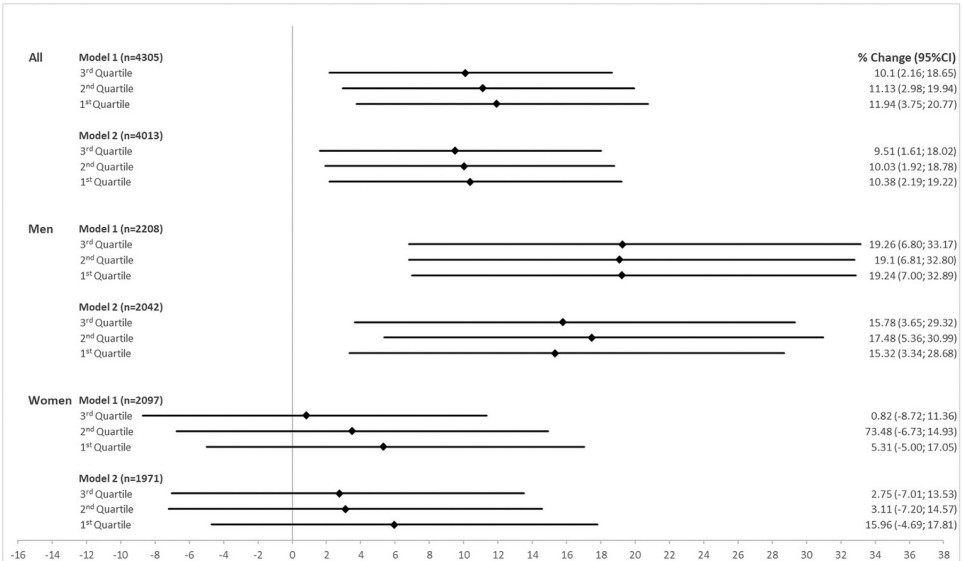

**Fig 2. Effect size estimates as percentage change in NT-proBNP and 95% confidence intervals (95%-CI) using sex-specific income quartiles (4th quartile as reference) in the main analysis population and stratified by sex.**

the observed sex-differences in the strength of the association between SEP indicators and NT-proBNP. There may be other factors affecting NT-proBNP not included in the present study, which are more prevalent in men, explaining the stronger effects of SEP indicators observed. Results suggest that prevention strategies for CVD aiming at those traditional CVD risk factors included in the present analysis may not led to a reduction of SEP-related differences in NT-proBNP or heart failure, respectively.

After excluding study participants with prevalent coronary heart disease or stroke, sensitivity analysis yielded similar results compared to the main analysis population. Although effect size estimates were slightly smaller in the sensitivity analysis, results did not indicate that the

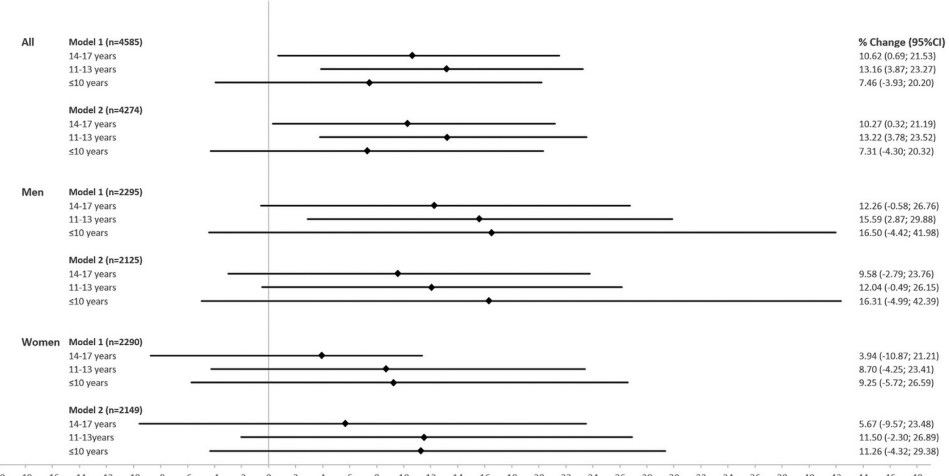

**Fig 3. Effect size estimates as percentage change in NT-proBNP and 95% confidence intervals (95%-CI) using education categories (≥18 years of education as reference) in the main analysis population and stratified by sex.**

observed associations between SEP and NT-proBNP were strongly influenced by overt coronary heart disease or stroke more prevalent in groups of low SEP.

The two SEP indicators included in the analysis represent different causal mechanisms. Income is discussed as to enable the individual investment in health-promoting goods and thus to directly influence living and environmental conditions [30, 31]. As effect size estimates were stronger for income as SEP indicator, material factors may play a bigger part in explaining inequalities in NT-proBNP plasma concentration compared to factors related to education, which is discussed to allow for the acquisition of positive social, psychosocial and economic resources having an impact on cardiovascular fitness [29, 32, 33]. Sex differences in the strength of associations between income and disease may also reflect income differences between men and women. In the present study household income was used as SEP indicator. The household income distribution between sexes only partly reflects sex-related SEP differences in the individual income common in almost all societies with women reporting a lower average individual income compared to men [34]. However, while it is generally assumed that using household income for investigating SEP-differences implies equal access to pooled resources for both sexes, this is in many populations not the case [34]. It could be hypothesized that the variance of the income spend for health-promoting goods may be higher in men, leading to more pronounced income-related health differences as observed in the present study for the association between household income and NT-proBNP in men.

Strength of the present study were its large population-based sample and the availability of cardiovascular risk factors for analysis, allowing for sex-stratified and mediation analysis. However, a limitation of the study was the cross-sectional analysis design. Individuals who have already died from a cardiovascular event were not included in the study population a priori and thus were missing in the distribution of NT-proBNP. However, by excluding participants with prevalent coronary heart disease or stroke at baseline we were able to demonstrate, that the association of SEP and NT-proBNP was still indicated in a sample free of these conditions. Another limitation was that no valid information on heart failure diagnosis at study baseline was available for analysis.

In the present study, associations of SEP indicators with NT-proBNP were demonstrated in a population-based cohort with differences in the strength of the associations between sexes. In future research, these findings should be investigated in prospective studies to shed more light on explanatory approaches and potential mechanisms for the associations observed.

## Supporting information

**S1 Table. Characteristics of the sensitivity analysis study population (i.e., excluding participants with prevalent coronary heart disease and stroke) stratified by sex (mv: Number of missing values).**
(DOCX)

**S2 Table. Effect size estimates as percentage change in NT-proBNP per 1000€ income/ month and 95% confidence intervals (95%-CI) for the main analysis population and stratified by sex.**
(DOCX)

**S3 Table. Effect size estimates as percentage change in NT-proBNP per 1000€ income/ month and 95% confidence intervals (95%-CI) for the main analysis population adjusted for age, sex and separately for one single cardiovascular risk factor.**
(DOCX)

**S4 Table. Effect size estimates as percentage change in NT-proBNP per year of education and 95% confidence intervals (95%-CI) for the main analysis population and stratified by sex.**
(DOCX)

**S5 Table. Effect size estimates as percentage change in NT-proBNP per year of education and 95% confidence intervals (95%-CI) for the main analysis population adjusted for age, sex and separately for one single cardiovascular risk factor.**
(DOCX)

**S6 Table. Effect size estimates as percentage change in NT-proBNP per 1000€ income/ month and 95% confidence intervals (95%-CI) for the analysis population after excluding participants with prevalent coronary heart disease and stroke and stratified by sex.**
(DOCX)

**S7 Table. Effect size estimates as percentage change in NT-proBNP per year of education and 95% confidence intervals (95%-CI) for the analysis population after excluding participants with prevalent coronary heart disease and stroke and stratified by sex.**
(DOCX)

**S8 Table. Effect size estimates as percentage change in NT-proBNP and 95% confidence intervals (95%-CI) using sex-specific income quartiles (4th as reference) in the analysis population after excluding participants with prevalent coronary heart disease and stroke and stratified by sex.**
(DOCX)

**S9 Table. Effect size estimates as percentage change in NT-proBNP and 95% confidence intervals (95%-CI) using education categories ($\geq$18 years of education as reference) in the analysis population after excluding participants with prevalent coronary heart disease and stroke and stratified by sex.**
(DOCX)

## Acknowledgments

We are indebted to all study participants and to both the dedicated personnel of the study center of the Heinz Nixdorf Recall study and to the investigative group, in particular to U. Slomiany, U. Roggenbuck, E. M. Beck, A. Öffner, S. Münkel, R. Peter, and H. Hirche.

Advisory Board: Meinertz T., Hamburg, Germany (Chair); Bode C., Freiburg, Germany; deFeyter P. J., Rotterdam, Netherlands; Güntert B, Halli, Austria; Gutzwiller F., Bern, Switzerland; Heinen H., Bonn, Germany; Hess O., Bern, Switzerland; Klein B., Essen, Germany; Löwel H., Neuherberg, Germany; Reiser M., Munich, Germany; Schmidt G., Essen, Germany; Schwaiger M., Munich, Germany; Steinmüller C., Bonn, Germany; Theorell T., Stockholm, Sweden; Willich S. N., Berlin, Germany.

## Author Contributions

**Conceptualization:** Börge Schmidt.

**Data curation:** Kaffer Kara, Amir Abbas Mahabadi, Raimund Erbel, Karl-Heinz Jöckel, Nico Dragano.

**Formal analysis:** Marina Rudman, Emanuel Matusch, Börge Schmidt.

**Funding acquisition:** Raimund Erbel, Karl-Heinz Jöckel.

**Investigation:** Marina Rudman, Mirjam Frank, Carina Emmel, Emanuel Matusch, Kaffer Kara, Amir Abbas Mahabadi, Raimund Erbel, Karl-Heinz Jöckel, Nico Dragano, Börge Schmidt.

**Methodology:** Nico Dragano, Börge Schmidt.

**Project administration:** Börge Schmidt.

**Supervision:** Börge Schmidt.

**Validation:** Mirjam Frank, Carina Emmel, Börge Schmidt.

**Visualization:** Mirjam Frank.

**Writing – original draft:** Marina Rudman, Börge Schmidt.

**Writing – review & editing:** Marina Rudman, Mirjam Frank, Carina Emmel, Emanuel Matusch, Kaffer Kara, Amir Abbas Mahabadi, Raimund Erbel, Karl-Heinz Jöckel, Nico Dragano, Börge Schmidt.

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
