## [Decision Letter · Decision Letter 0]

19 Apr 2021

PONE-D-20-33242

Social Inequalities in N-Terminal pro‑Brain Natriuretic Peptide (NT‑proBNP) – Results of the Population‑based Heinz Nixdorf Recall Study

PLOS ONE

Dear Dr. Schmidt,

Thank you for submitting your manuscript to PLOS ONE. After careful consideration, we feel that it has merit but does not fully meet PLOS ONE’s publication criteria as it currently stands. Therefore, we invite you to submit a revised version of the manuscript that addresses the points raised during the review process.

Please address all comments raised by the two reviewers, in particular regarding additional methodological details, statistical analysis, and an explanation of your results.

We look forward to receiving your revised manuscript.

Sincerely,

Yann Benetreau, Ph.D.

Senior Editor, *PLOS ONE*

Journal Requirements:

PLOS requires an ORCID iD for the corresponding author in Editorial Manager on papers submitted after December 6th, 2016. Please ensure that you have an ORCID iD and that it is validated in Editorial Manager. To do this, go to ‘Update my Information’ (in the upper left-hand corner of the main menu), and click on the Fetch/Validate link next to the ORCID field. This will take you to the ORCID site and allow you to create a new iD or authenticate a pre-existing iD in Editorial Manager. Please see the following video for instructions on linking an ORCID iD to your Editorial Manager account: https://www.youtube.com/watch?v=_xcclfuvtxQ

In statistical methods, please clarify whether you corrected for multiple comparisons.

In your statistical analyses, please state whether you accounted for survey weights and clustering by region - did you consider using multilevel models?

5a) If there are ethical or legal restrictions on sharing a de-identified data set, please explain them in detail (e.g., data contain potentially identifying or sensitive patient information) and who has imposed them (e.g., an ethics committee). Please also provide contact information for a data access committee, ethics committee, or other institutional body to which data requests may be sent.

5b) If there are no restrictions, please upload the minimal anonymized data set necessary to replicate your study findings as either Supporting Information files or to a stable, public repository and provide us with the relevant URLs, DOIs, or accession numbers. Please see http://www.bmj.com/content/340/bmj.c181.long for guidelines on how to de-identify and prepare clinical data for publication. For a list of acceptable repositories, please see http://journals.plos.org/plosone/s/data-availability#loc-recommended-repositories.

Reviewers' comments:

Reviewer's Responses to Questions

**Comments to the Author**

1. Is the manuscript technically sound, and do the data support the conclusions?

Reviewer #1: Yes

Reviewer #2: No

2. Has the statistical analysis been performed appropriately and rigorously? 

Reviewer #1: Yes

Reviewer #2: No

3. Have the authors made all data underlying the findings in their manuscript fully available?

Reviewer #1: Yes

Reviewer #2: No

4. Is the manuscript presented in an intelligible fashion and written in standard English?

Reviewer #1: Yes

Reviewer #2: Yes

5. Review Comments to the Author

Reviewer #1: Thanks for the possibility to review the manuscript: “Social Inequalities in N-Terminal pro-Brain Natriuretic Peptide (NT-proBNP) – Results of the Population-based Heinz Nixdorf Recall Study. It is an interesting epidemiological study of a population based sample of 45-75 year old men and women, studying the association of NT-proBNP level in men and women and educational and income level.

I think it is well written and interesting, but I have some major suggestions for modifications of the manuscript. I also would like the authors to add a discussion of clinical relevance to the manuscript.

Here follows my comments:

Title and abstract

I think it is easy to misunderstand the concept social inequality in the title and the conclusions (abstract and main manuscript). I would prefer disparity or diversity - it is a bit hard to accept that the level of NTproBNP could be a social inequality.

Background

The complex gender differences in CVD prevalences concerning IHD and heart failure (HF) is not illustrated in the background part of the manuscript. .As there is a considerable difference in prevalence of IHD and HF in men compared to women in different ages, this should be highlighted, (both concerning IHD and HF).

Basically, NT-proBNP is a marker for HF, and a normal NT-proBNP can exclude HF, but not IHD, and its importance in the diagnosis of HF is the high negative predictive value – a level <300 pg/mL excludes HF. Internationally a level < 125 pg/mL is said to be normal. This has relevance in the discussion concerning adding NT-proBNP as yet another risk factor for CVD and especially concerning gender differences, as mean NT-proBNP levels in women are higher than in men.

In the sentence ... via unequally distributed risk factors such as health-related behaviors (e.g., smoking, nutrition), psychosocial factors (e.g., stress) and different material factors (e.g., living, working and housing conditions) … I suppose “stress” means mental stress .

In the next part of Background where you present the following: “Especially in low SEP groups a substantial proportion of individuals may be undertreated while showing subclinical signs of being at high risk for developing CVD events”, known sex differences concerning low SEP as a risk factor should be identified as well as including prevention/health promotion besides treatment as an important concept.

Methods

Why is only individuals with CHD and stroke excluded? Why not HF?

Could you choose another wording than “strong” in” No strong differences in the distribution of traditional CVD risk factors were observed”?

Results

It is striking that although women have a mean NTproBNP higher than men, all other risk factors are lower, HDL is higher. Besides this, lipid lowering frequency is higher although diabetes has an almost doubled prevalence in men. The difference between men and women is troubling, and although you don´t present statistical testing of differences between men and women I think you need to present the important divergent directions of risk factors and mean values in men compared to women in table 1 to make the reader observant of this fact .

You exclude prevalent CHD and stroke from several of the analyses (S4, S5) but not HF?

The authors claim they excluded persons with CVD in some analyses, which is not quite adequate, as only persons with CHD and stroke (HF?) were excluded, but not persons with hypertension. I understand why you don´t include hypertension, but this diagnosis is included in CVD.

Discussion

In discussion I think it is important also to include the clinical relevance of the findings, as well as the apparent differences between men and women both concerning incidence, prevalence and, also very important, prevention. For example, the apparent difference of frequency of lipid lowering medication in women and men, although the considerably higher diabetes prevalence in men is interesting in this respect, and could be included in the discussion about prevention.

In Discussion the authors claim “ As effect size estimates were stronger for income as SEP indicator, material factors may play a bigger part in explaining inequalities in NT-proBNP plasma concentration compared to factors related to education, which is discussed to allow for the acquisition of positive social, psychosocial and economic resources having an impact on cardiovascular fitness.” Given the great discrepancies between men and women, with women showing higher NTporBNP although less risk increase and great differences between income and education compared to men, I think a complementary discussion concerning why income could explain differences for women should be included.

Reviewer #2: PONE-D-20-33242: statistical review

SUMMARY This is a cross-sectional study that investigates whether income and/or educational levels are associated with individual levels of NT‑proBNP, a diagnostic marker for heart failure and a prognostic factor for cardiovascular disease. The statistical analysis relies on a battery of linear regression models, which are estimated by separately including income and education. Although the results seem sound, I have several concerns about the methods.

MAJOR ISSUES

1. Both income and education have been used as continuous variables or divided into four groups using sex-specific quartiles. Categorization of a continuous variable is an unnecessary waste of information and it is generally not recommended. In addition, results arbitrarily depend on the cut-points that have been chosen. What is the motivation for this categorization?

2. The regression coefficients of the linear regression analysis have been transformed and presented as percentage changes. These estimates are displayed along with confidence intervals. What methods has been chosen to compute the standard errors of the transformed coefficients?

3. Figure 1 seems to indicate some data heteroscedasticity. Did the authors either check or account for heteroscedasticity in the residuals of the linear regression models? Assuming homoscedastic errors in the presence of heteroscedasticity could lead to wrong standard errors.

4. Tables 2 and 3 display a subset of the regression output. All model estimates should be however provided, including the effects of the confounders. This is not only for the sake of clarity: it will also guarantee results reproducibility.

5. Model checking is overlooked. Could the authors provide evidence of the normality of the dependent variable and the goodness of fit of the estimated models?

Typo:

page 15 "The two SEP indicators included in the analysis representing different causal mechanisms", the sentence doesn't sound right ...

6. PLOS authors have the option to publish the peer review history of their article (what does this mean?). If published, this will include your full peer review and any attached files.

Reviewer #1: No

Reviewer #2: No

---

## [Author Response · Author response to Decision Letter 0]

8 Jun 2021

PONE-D-20-33242

Social Inequalities in N-Terminal pro Brain Natriuretic Peptide (NT proBNP) – Results of the Population based Heinz Nixdorf Recall Study

We would like to thank both reviewers and the editor for the very helpful comments. We have carefully addressed each comment within the revised article and/or in our specific responses below. All changes to the manuscript are described in text below, each response, additions and deletions are shown in the revised manuscript using tracked changes. We believe that these changes have improved the manuscript, which we hope is suitable for publication now.

Reviewer #1: Thanks for the possibility to review the manuscript: “Social Inequalities in N-Terminal pro-Brain Natriuretic Peptide (NT-proBNP) – Results of the Population-based Heinz Nixdorf Recall Study. It is an interesting epidemiological study of a population based sample of 45-75 year old men and women, studying the association of NT-proBNP level in men and women and educational and income level. I think it is well written and interesting, but I have some major suggestions for modifications of the manuscript. I also would like the authors to add a discussion of clinical relevance to the manuscript.

Here follows my comments: 

Title and abstract

I think it is easy to misunderstand the concept social inequality in the title and the conclusions (abstract and main manuscript). I would prefer disparity or diversity - it is a bit hard to accept that the level of NTproBNP could be a social inequality.

The title was based on the phrase “Social Inequalities in Health and Disease” frequently used if the incidence of a disease (or a marker of disease) is unequally distributed across social groups. However, we changed the title now to “Socioeconomic Position is Associated with N-Terminal pro-Brain Natriuretic Peptide (NT-proBNP) – Results of the Population-based Heinz Nixdorf Recall Study” and accordingly the phrasing in abstract and main text. 

Background

The complex gender differences in CVD prevalences concerning IHD and heart failure (HF) is not illustrated in the background part of the manuscript. .As there is a considerable difference in prevalence of IHD and HF in men compared to women in different ages, this should be highlighted, (both concerning IHD and HF). Basically, NT-proBNP is a marker for HF, and a normal NT-proBNP can exclude HF, but not IHD, and its importance in the diagnosis of HF is the high negative predictive value – a level <300 pg/mL excludes HF. Internationally a level < 125 pg/mL is said to be normal. This has relevance in the discussion concerning adding NT-proBNP as yet another risk factor for CVD and especially concerning gender differences, as mean NT-proBNP levels in women are higher than in men.

We thank reviewer 1 for this advice. We have now added this topic to the introduction (please see page 5 of the revised manuscript using track changes).

In the sentence ... via unequally distributed risk factors such as health-related behaviors (e.g., smoking, nutrition), psychosocial factors (e.g., stress) and different material factors (e.g., living, working and housing conditions) … I suppose “stress” means mental stress .

Yes, we have now added this information to the respective sentence (please see page 5 of the revised manuscript using track changes).

In the next part of Background where you present the following: “Especially in low SEP groups a substantial proportion of individuals may be undertreated while showing subclinical signs of being at high risk for developing CVD events”, known sex differences concerning low SEP as a risk factor should be identified as well as including prevention/health promotion besides treatment as an important concept.

We have now added this topic at the end of the first paragraph of the introduction (please see page 5 of the revised manuscript using track changes).

Methods

Why is only individuals with CHD and stroke excluded? Why not HF?

Information on validated HF diagnosis at study baseline was not available for the study population. As heart failure (for instance, compared to myocardial infarction) is a more “soft” endpoint, information on self-reported diagnosis of heart failure is not as valid as for harder endpoints. We have added this to the limitation section (please see page 18 of the revised manuscript using track changes).

Could you choose another wording than “strong” in” No strong differences in the distribution of traditional CVD risk factors were observed”?

We now have used the word “substantial” (please see page 10 of the revised manuscript using track changes).

Results

It is striking that although women have a mean NTproBNP higher than men, all other risk factors are lower, HDL is higher. Besides this, lipid lowering frequency is higher although diabetes has an almost doubled prevalence in men. The difference between men and women is troubling, and although you don´t present statistical testing of differences between men and women I think you need to present the important divergent directions of risk factors and mean values in men compared to women in table 1 to make the reader observant of this fact .

We have now described the divergent distribution of risk factors between men and women in more detail at the beginning of the results section (please see page 10 of the revised manuscript using track changes).

You exclude prevalent CHD and stroke from several of the analyses (S4, S5) but not HF?

The authors claim they excluded persons with CVD in some analyses, which is not quite adequate, as only persons with CHD and stroke (HF?) were excluded, but not persons with hypertension. I understand why you don´t include hypertension, but this diagnosis is included in CVD.

We have now corrected this and consistently wrote “coronary heart disease or stroke” instead of “CVD” throughout the manuscript. With regard to heart failure please see our response above.

Discussion

In discussion I think it is important also to include the clinical relevance of the findings, as well as the apparent differences between men and women both concerning incidence, prevalence and, also very important, prevention. For example, the apparent difference of frequency of lipid lowering medication in women and men, although the considerably higher diabetes prevalence in men is interesting in this respect, and could be included in the discussion about prevention.

We have now added the topic of prevention, especially with respect to the reduction of inequalities in heart failure to the 3rd paragraph of the discussion section (please see page 17 of the revised manuscript using track changes). 

In Discussion the authors claim “ As effect size estimates were stronger for income as SEP indicator, material factors may play a bigger part in explaining inequalities in NT-proBNP plasma concentration compared to factors related to education, which is discussed to allow for the acquisition of positive social, psychosocial and economic resources having an impact on cardiovascular fitness.” Given the great discrepancies between men and women, with women showing higher NTporBNP although less risk increase and great differences between income and education compared to men, I think a complementary discussion concerning why income could explain differences for women should be included.

We have now extended the discussion according to reviewer 1 suggestions (please see page 18 of the revised manuscript using track changes).

Reviewer #2: PONE-D-20-33242: statistical review

SUMMARY This is a cross-sectional study that investigates whether income and/or educational levels are associated with individual levels of NT proBNP, a diagnostic marker for heart failure and a prognostic factor for cardiovascular disease. The statistical analysis relies on a battery of linear regression models, which are estimated by separately including income and education. Although the results seem sound, I have several concerns about the methods.

MAJOR ISSUES

1. Both income and education have been used as continuous variables or divided into four groups using sex-specific quartiles. Categorization of a continuous variable is an unnecessary waste of information and it is generally not recommended. In addition, results arbitrarily depend on the cut-points that have been chosen. What is the motivation for this categorization?

In the present study both income and education were not assessed as metric variables. Education has been assessed asking study participants for their highest degrees in formal education. Then this information has been transformed according to the International Standard Classification of Education (ISCED) to get a continuous measure of the total years of formal education necessary to attain the reported educational degree in Germany. Using ISCED is recommended to make information on education internationally comparable. We have used ISCED years of total education as continuous measure in order to avoid the reduction of information, but have additionally used education categories that reflect different classes of educational attainment in Germany to check for the robustness of our results. Thus, education categories were not arbitrary chosen for our analysis and were used in the same way in all previous analyses in the Heinz Nixdorf Recall Study when education was the main exposure of interest (e.g., Dragano N, Verde PE, Moebus S, Stang A, Schmermund A, Roggenbuck U, et al. Subclinical coronary atherosclerosis is more pronounced in men and women with lower socioeconomic status. Associations in a population based study. Eur J Cardiovasc Prev Rehabil. 2007;14(4):568-74.).

Household income was also assed using predefined income categories, but then used as continuous as well as categorical variable, also to check for the robustness of results.

As both approaches – using SEP indicators as continuous AND categorical variables – led to very comparable results and identical conclusions, we were able to demonstrate the robustness of our results, which do not seem to be strongly dependent on the way SEP indicators were used in the analysis. We have now added the motivation for analyzing both variables in the methods section (please see page 9 of the revised manuscript using track changes).

2. The regression coefficients of the linear regression analysis have been transformed and presented as percentage changes. These estimates are displayed along with confidence intervals. What methods has been chosen to compute the standard errors of the transformed coefficients?

Residual standard errors of the standard lm function in R were used to compute the 95% confidence intervals.

3. Figure 1 seems to indicate some data heteroscedasticity. Did the authors either check or account for heteroscedasticity in the residuals of the linear regression models? Assuming homoscedastic errors in the presence of heteroscedasticity could lead to wrong standard errors.

To check the impact of potential heteroscedasticity on the main study results we re-calculated standard errors for the main models (i.e., models presented in tables 2 and 3) using HC3 robust covariance matrix estimators as implemented in the sandwich package in R (https://cran.r-project.org/web/packages/sandwich/sandwich.pdf). The re-calculated standard errors differed only very slightly compared to the residual standard errors that were initially calculated assuming homoscedasticity. This indicates that potential heteroscedasticity does not have to seem a relevant impact on the study results.

E.g. for the main results indicating that income is associated with log(NT-proBNP) in the overall study population, the male and the female subgroup:

Output of the standard linear regression function in r for model 1, All, table 2:

 Estimate Std. Error t value Pr(>|t|) 

(Intercept) 1.560e+00 1.119e-01 13.938 < 2e-16

income_EUR -6.687e-05 1.908e-05 -3.504 0.000463

sex 3.742e-01 2.680e-02 13.962 < 2e-16

age 4.460e-02 1.706e-03 26.145 < 2e-16

Output for the same model using the R sandwich package to calculate robust standard errors:

 Estimate Std. Error t value Pr(>|t|) 

(Intercept) 1.5596e+00 1.1205e-01 13.9189 < 2.2e-16

income_EUR -6.6868e-05 1.8823e-05 -3.5525 0.0003857

sex 3.7416e-01 2.6629e-02 14.0507 < 2.2e-16

age 4.4599e-02 1.7429e-03 25.5894 < 2.2e-16

##############

Output of the standard linear regression function in r for model 1, Men, table 2:

 Estimate Std. Error t value Pr(>|t|) 

(Intercept) 8.092e-01 1.620e-01 4.995 6.36e-07

income_EUR -8.795e-05 2.737e-05 -3.213 0.00133 

age 5.777e-02 2.516e-03 22.964 < 2e-16

Output for the same model using the R sandwich package to calculate robust standard errors:

 Estimate Std. Error t value Pr(>|t|) 

(Intercept) 8.0923e-01 1.6186e-01 4.9995 6.201e-07

income_EUR_M -8.7947e-05 2.6583e-05 -3.3084 0.0009535

age 5.7766e-02 2.5619e-03 22.5484 < 2.2e-16

##############

Output of the standard linear regression function in r for model 1, Women, table 2:

 Estimate Std. Error t value Pr(>|t|) 

(Intercept) 2.720e+00 1.465e-01 18.563 <2e-16

income_EUR -5.230e-05 2.601e-05 -2.011 0.0445 

age 3.102e-02 2.255e-03 13.755 <2e-16

Output for the same model using the R sandwich package to calculate robust standard errors:

 Estimate Std. Error t value Pr(>|t|) 

(Intercept) 2.71967524 0.14650616 18.5636 <2e-16

income_EUR_F -0.00005230 0.00002623 -1.9939 0.0463 

age 0.03102485 0.00228165 13.5976 <2e-16

##############

The results given above for our main findings also show that statistical significance at α = 0.05 was not affected by the very small difference between the different standard errors. Based on this sensitivity analysis we decided to present the original results to avoid doubling the overall number of regression models calculated for this study. We hope that reviewer 2 can agree with this.

4. Tables 2 and 3 display a subset of the regression output. All model estimates should be however provided, including the effects of the confounders. This is not only for the sake of clarity: it will also guarantee results reproducibility.

We have now presented all effect size estimators for Table 2 and 3 in the supplement (please see tables S2 und S4 in the supplement).

5. Model checking is overlooked. Could the authors provide evidence of the normality of the dependent variable and the goodness of fit of the estimated models?

NT-proBNP was not normally distributed in the study population (s. histogram below):

By using the natural logarithm of NT-proBNP the distribution was transformed sufficiently to assume normality for including it in a linear regression analysis with a large N of ~4600 observations (s. histogram below):

We have now added additional information in the methods section to explain the rationale behind the log-transformation of NT-proBNP (please see page 8 of the revised manuscript using track changes).

The Q-Q plot below for the residuals of the association between income and log(NT-proBNP) for the overall study population using model 1 gave indication for sufficient Goodness of Fit with some indication that for high values of log(NT-proBNP), the fitted values are slightly smaller than the actual values:

The total R-squared for the association between income and log(NT-proBNP) using model 1 was 0.18. Using model 2 (i.e., additionally including CVR risk factors) R-squared was 0.23. However, the main aim of the study was not to improve the prediction of NT-proBNP and getting a large R-squared, but to establish the association between SEP indicators and NT-proBNP, for which previous studies already found evidence (7. Vart P, Matsushita K, Rawlings AM, Selvin E, Crews DC, Ndumele CE, Coresh J. SEP, Heart Failure, and N-terminal Pro-b-type Natriuretic Peptide: The Atherosclerosis Risk in Communities Study. Am J Prev Med. 2018;54(2):229–36).

We strongly believe that our results (including descriptive analysis, association analyses with several sensitivity analyses) are sound and demonstrate the robustness of our findings. The additional calculations requested by Reviewer 2 were very helpful for strengthening our study and we hope that we could eliminate Reviewer 2’s methodological concerns. 

Typo:

page 15 "The two SEP indicators included in the analysis representing different causal mechanisms", the sentence doesn't sound right ...

We have now corrected this sentence (“The two SEP indicators included in the analysis represent different causal mechanisms", please see page 18 of the revised manuscript using track changes).

---

## [Decision Letter · Decision Letter 1]

26 Jul 2021

Socioeconomic Position is Associated with N-Terminal pro‑Brain Natriuretic Peptide (NT‑proBNP) – Results of the Population‑based Heinz Nixdorf Recall Study

PONE-D-20-33242R1

Dear Dr. Schmidt,

We’re pleased to inform you that your manuscript has been judged scientifically suitable for publication and will be formally accepted for publication once it meets all outstanding technical requirements.

Kind regards,

Gualtiero I. Colombo, M.D., Ph.D.

Academic Editor

PLOS ONE

Additional Editor Comments (optional):

Reviewers' comments:

Reviewer's Responses to Questions

**Comments to the Author**

1. If the authors have adequately addressed your comments raised in a previous round of review and you feel that this manuscript is now acceptable for publication, you may indicate that here to bypass the “Comments to the Author” section, enter your conflict of interest statement in the “Confidential to Editor” section, and submit your "Accept" recommendation.

Reviewer #2: All comments have been addressed

2. Is the manuscript technically sound, and do the data support the conclusions?

Reviewer #2: (No Response)

3. Has the statistical analysis been performed appropriately and rigorously? 

Reviewer #2: (No Response)

4. Have the authors made all data underlying the findings in their manuscript fully available?

Reviewer #2: (No Response)

5. Is the manuscript presented in an intelligible fashion and written in standard English?

Reviewer #2: (No Response)

6. Review Comments to the Author

Reviewer #2: (No Response)

7. PLOS authors have the option to publish the peer review history of their article (what does this mean?). If published, this will include your full peer review and any attached files.

Reviewer #2: No

---

## [Editor Report · Acceptance letter]

12 Aug 2021

PONE-D-20-33242R1 

Socioeconomic Position is Associated with *N-Terminal pro‑Brain Natriuretic Peptide (NT‑proBNP)* – Results of the Population‑based Heinz Nixdorf Recall Study 

Dear Dr. Schmidt:

I'm pleased to inform you that your manuscript has been deemed suitable for publication in PLOS ONE. Congratulations! Your manuscript is now with our production department. 

Kind regards, 

on behalf of

Dr. Gualtiero I. Colombo 

Academic Editor

PLOS ONE